# In Vitro Cytotoxicity and Antimicrobial Activity against Acne-Causing Bacteria and Phytochemical Analysis of Galangal (*Alpinia galanga*) and Bitter Ginger (*Zingiber zerumbet*) Extracts

**DOI:** 10.3390/ijms252010869

**Published:** 2024-10-10

**Authors:** Tanat Na Nongkhai, Sarah E. Maddocks, Santi Phosri, Sarita Sangthong, Punyawatt Pintathong, Phanuphong Chaiwut, Kasemsiri Chandarajoti, Lutfun Nahar, Satyajit D. Sarker, Tinnakorn Theansungnoen

**Affiliations:** 1School of Cosmetic Science, Mae Fah Luang University, Chiang Rai 57100, Thailand; 6551704001@lamduan.mfu.ac.th (T.N.N.); sarita.san@mfu.ac.th (S.S.); punyawatt.pin@mfu.ac.th (P.P.); phanuphong@mfu.ac.th (P.C.); 2Green Cosmetic Technology Research Group, School of Cosmetic Science, Mae Fah Luang University, Chiang Rai 57100, Thailand; 3Microbiology and Infection Research Group, Cardiff School of Sport and Health Sciences, Cardiff Metropolitan University, Llandaff, Cardiff CF5 2YB, UK; smaddocks@cardiffmet.ac.uk; 4Department of Chemical Engineering, Faculty of Engineering, Burapha University, Chonburi 20131, Thailand; santi.ph@eng.buu.ac.th; 5Department of Pharmaceutical Chemistry, Faculty of Pharmaceutical Sciences, Prince of Songkla University, Hat-Yai, Songkhla 90112, Thailand; kasemsiri.c@psu.ac.th; 6Drug Delivery System Excellence Center, Faculty of Pharmaceutical Sciences, Prince of Songkla University, Hat-Yai, Songkhla 90112, Thailand; 7Laboratory of Growth Regulators, Palacký University and Institute of Experimental Botany, The Czech Academy of Sciences, Šlechtitelů 27, 78371 Olomouc, Czech Republic; nahar@ueb.cas.cz; 8Centre for Natural Products Discovery, School of Pharmacy and Biomolecular Sciences, Liverpool John Moores University, Byrom Street, Liverpool L3 3AF, UK; s.sarker@ljmu.ac.uk

**Keywords:** *Acne vulgaris*, *Alpinia galanga*, antimicrobial, cytotoxicity, phytochemicals, *Zingiber zerumbet*

## Abstract

Galangal (*Alpinia galanga* (L.) Willd) and bitter ginger (*Zingiber zerumbet* (L.) Roscoe) are aromatic rhizomatous plants that are typically used for culinary purposes. These rhizomatous plants have many biological properties and the potential to be beneficial for pharmaceutics. In this study, we evaluated the antioxidant and antimicrobial activities, with a specific focus on acne-causing bacteria, as well as the phytochemical constituents, of different parts of galangal and bitter ginger. The rhizomes, stems, and leaves of galangal and bitter ginger were separately dried for absolute ethanol and methanol extractions. The extracts were used to evaluate the antioxidant activity using a DPPH radical scavenging assay (0.005–5000 μg/mL), antimicrobial activity against acne-causing bacteria (0.50–31.68 mg/mL), and in vitro cytotoxicity toward human keratinocytes and fibroblasts (62.5–1000 μg/mL), as well as analyses of bioactive phytochemicals via GC-MS and LC-MS/MS (500 ppm). The ethanol and methanol extracts of bitter ginger and galangal’s rhizomes (BRhE, BRhM, GRhE, and GRhM), stems (BStE, BStM, GRhE, and GRhM), and leaves (BLeE, BLeM, GLeE, and GLeM), respectively, showed antioxidant and antimicrobial activities. The extracts of all parts of bitter ginger and galangal were greatly antioxidative with 0.06–1.42 mg/mL for the IC_50_ values, while most of the extracts were strongly antimicrobial against *C. acnes* DMST 14916, particularly BRhM, BRhE, GRhM, and GRhE (MICs: 3.96–7.92 mg/mL). These rhizome extracts had also antimicrobial activities against *S. aureus* TISTR 746 (MICs: 7.92–31.68 mg/mL) and *S. epidermidis* TISTR 518 (MICs: 7.92–15.84 mg/mL). The extracts of bitter ginger and galangal rhizomes were not toxic to HaCaT and MRC-5 even at the highest concentrations. Through GC-MS and LC-MS/MS analysis, phytochemicals in bitter ginger rhizome extracts, including zerumbone, tectorigenin, piperic acid, demethoxycurcumin, and cirsimaritin, and galangal rhizome extracts, including sweroside and neobavaisoflavone, were expected to provide the antioxidant and anti-microbial activities. Therefore, the results suggest that the bitter ginger and galangal extracts could be natural anti-acne compounds with potential for pharmaceutic, cosmetic, and aesthetic applications.

## 1. Introduction

Pathogenic bacteria are responsible for diverse infectious and chronic health conditions with severe impacts on quality of life. The infection process relies on bacterial adaptation to the various protective and immune components of the human body, including physical barriers, such as the skin. However, infection of the skin can occur when bacteria enter through damaged skin. Such infections are both common and mild, including erythema, edema, and localized inflammation [1]. Acne (*Acne vulgaris*) is a common inflammatory skin disorder, which is the result of infection and the colonization of pilosebaceous follicles by the anaerobic Gram-positive bacterium *Cutibacterium acnes*. Bacterial colonization results in blockage and/or inflammation of the pilosebaceous follicles, with acne usually affecting adolescents (>85%), but sometimes persisting into adulthood [2]. Acne can adversely affect life quality of the people in both physiological and psychosocial ways. Regarding the physiological impacts, acne can cause an abnormal skin appearance, such as acne lesions with different severities and acne scars. Regarding the psychosocial impacts, acne causes negative effects on self-esteem leading to feelings of social isolation and loneliness [3].

The pathogenesis of acne involves four important factors, including the overproduction of sebum, hyperkeratinization of pilosebaceous follicles, hyperproliferation of *C. acnes* (formerly *Propionibacterium acnes*), and inflammation [4]. *C. acnes* is not only a common cause of acne but also a crucial factor for its progression and severity [5,6]. In severe acne, *C. acnes* and *Staphylococcus epidermidis* are reported to promote pus formation and lead to inflammatory acne lesions [6,7]. Furthermore, *S. epidermidis* and the known pathogen *Staphylococcus aureus* are routinely co-isolated from acne patients, and the prevalence is independent of gender [8]. From a previous study, acne patients had levels of oxidants, including malondialdehyde and nitric oxide, that were increased significantly, while their activities of superoxide dismutase and catalase were decreased significantly, compared to control subjects. The results suggested that the antioxidant defense system of the acne patients is dysfunctional, and therefore, antioxidants can be indicated for acne treatment [9].

The appropriate treatments for acne vulgaris are typically considered depending on the types of acne and the severity levels of acne lesions [10,11]. For non-inflammatory or comedogenic acnes with mild to moderate levels, treatments with topical agents, including retinoids, azelaic acid, and benzoyl peroxide, are recommended, while for inflammatory or papulopustular acnes, treatments with a combination of the topical agents and antibiotics, such as benzoyl peroxide and clindamycin, are strongly recommended [10]. However, certain antibiotics have been found to induce bacterial resistance and even lead to the therapeutic failure of acne treatments [12,13]. To deal with the resistance and failure problems, novel antimicrobial agents derived from natural sources, in particular herbs and plants, are attempted to discover and develop as an alternative approach [14]. Moreover, studying the use of plants and phytochemicals in the treatment of acne vulgaris has recently emerged in different research [15].

*Alpinia galanga*, commonly called galangal, is an aromatic and herbal plant belonging to the family Zingiberaceae. The galangal rhizome is generally used as a spice and widely grown in many Asian countries, including Indonesia, Sri Lanka, India, Saudi Arabia, China [16], and even Thailand. The flowers and young shoots of galangal are also used as a spice or as a vegetable. The plant is broadly used in the traditional medication systems, such as Chinese, Ayurveda, Unani, and Thai folk medicine, and to treat many human diseases, such as inflammation, rheumatic pains, chest pain, diabetes, fever, kidney disease, and tumors [17,18]. Galangal is reported to contain several flavonoids and volatile oils and possesses many pharmacological and biological properties, including immunomodulatory, hypolipidemic, antidiabetic, antiplatelet, antioxidant, antiprotozoal, antiviral, antifungal, and antibacterial properties [18,19].

*Zingiber zerumbet* (L.) Roscoe ex Sm., known as shampoo ginger or bitter ginger, is a perennial and aromatic plant, which belongs to the family Zingiberaceae. Bitter ginger is widely grown in many Asian countries and is used for many beneficial purposes, including foods, beverages, and ornamental purposes [20]. The flowers of bitter ginger are cone-shaped and long-lasting and are used in craft arrangements for ornamental purposes, while the floral buds are commonly consumed as vegetables [20,21]. The rhizome of bitter ginger can be used as a food seasoning, tonic, and stimulant [20]. The plant rhizome has been also used in many traditional medicines, such as Indian, Thai, Chinese, and Arabic folkloric medicines [22]. The plant is a rich source of distinct classes of compounds, such as polyphenols, terpenes, and alkaloids [20]. Bitter ginger has a wide spectrum of pharmacological and biological properties, including carminative, diuretic, antipyretic, anti-diarrheal, antidiabetic, anti-inflammatory, and antibacterial properties [23].

This study sought to evaluate the effects of different solvents and plant parts on biological activities, including antioxidant and acne-causing antimicrobial activities, as well as the phytochemical constituents of galangal and bitter ginger. This was with a view of their application to manage acne vulgaris. To achieve this, ethanolic and methanolic extracts of leaves, stems, and rhizomes of galangal and bitter ginger were evaluated for antioxidant activity against DPPH radicals, antimicrobial activity against acne-causing bacteria (*C. acnes*, *S. aureus*, and *S. epidermidis*), and cytotoxic activity against human keratinocyte HaCaT and fibroblast MRC-5 cell lines. The phytochemical constituents of galangal and bitter ginger extracts were identified via GC-MS and LC-MS/MS analysis.

## 2. Results

### 2.1. Yields and Antioxidant Activity of Crude Extracts of Bitter Ginger and Galangal

Bitter ginger and galangal (Figure 1) were separated into three parts, including rhizomes, stems, and leaves, and taken to extraction. The yields of crude extracts of bitter ginger and galangal were obtained using ethanol and methanol extractions. The ethanol and methanol extracts of bitter ginger rhizomes (BRhE and BRhM), stems (BStE and BStM), and leaves (BLeE and BLeM), while those of galangal rhizomes (GRhE and GRhM), stems (GStE and GStM), and leaves (GLeE and GLeM), were obtained, respectively, and shown in Table 1. The yields of the rhizome extractions, including BRhE, BRhM, GRhE, and GRhM, were 5.17 ± 0.63%, 7.30 ± 0.09%, 5.47 ± 0.40%, and 6.94 ± 0.50%, respectively. The yields of the stem extractions, including BStE, BStM, GStE, and GStM, were 5.03 ± 0.76%, 1.06 ± 0.13%, 1.72 ± 0.06%, and 2.72 ± 0.62%, respectively. The yields of the leaf extractions, including BLeE, BleM, GLeE, and GLeM, were 2.14 ± 0.34%, 2.14 ± 0.37%, 5.37 ± 0.94%, and 5.67 ± 0.36%, respectively. Based on a statistical comparison between different solvents, the yields of galangal rhizomes extracted using ethanol and methanol were similar (5.47% and 6.94%). The yields of ethanolic and methanolic extracts of bitter ginger leaves (2.14% and 2.14%) and galangal leaves (5.37% and 5.67%), respectively, were not significantly different (*p >* 0.05). The ethanolic and methanolic extracts of bitter ginger rhizomes (5.71% and 7.30%) showed a significantly different yield (*p <* 0.05). The yields of ethanolic and methanolic extracts of bitter ginger stems (5.03% and 1.06%) and galangal stems (1.72% and 2.72%), respectively, were significantly different (*p <* 0.05).

The antioxidant activity of bitter ginger and galangal extracts was evaluated by performing a DPPH radical scavenging ability assay. The IC_50_ of ascorbic acid was 1.4 ± 0.2 μg/mL (a positive control). As the result shows in Table 1, the IC_50_ values of BRhE, BRhM, GRhE, and GRhM were 1.19 ± 0.06 mg/mL, 0.99 ± 0.04 mg/mL, 0.08 ± 0.01 mg/mL, and 0.06 ± 0.01 mg/mL, while those of BStE, BStM, GStE, and GStM were 1.42 ± 0.04 mg/mL, 0.46 ± 0.03 mg/mL, 0.15 ± 0.01 mg/mL, and 0.28 ± 0.02 mg/mL, respectively. The IC50 values of BLeE, BleM, GLeE, and GLeM were 0.40 ± 0.02 mg/mL, 0.30 ± 0.01 mg/mL, 0.27 ± 0.03 mg/mL, and 0.17 ± 0.02 mg/mL, respectively. After statistical analysis, the extraction of bitter ginger stems with methanol (BStM) resulted in more effective antioxidants than that with ethanol (BStE) (*p* < 0.05). The methanol extraction of galangal stems (GStM) was less effective to extract antioxidants than its ethanol extraction (GStE) (*p* < 0.05). The methanolic extracts of bitter ginger and galangal leaves (BLeM and GLeM) revealed higher antioxidant activity than their ethanolic extracts (BLeE and GLeE) (*p* < 0.05). The methanolic extract of bitter ginger rhizomes (BRhM) showed greater antioxidant activity than its ethanolic extract (BRhE) (*p* < 0.05), while the antioxidant activity of methanolic and ethanolic extracts of galangal rhizomes (GRhM and GRhE) was not significantly different (*p* < 0.05). Moreover, GRhM and GRhE showed the highest antioxidant activity among all extracts.

### 2.2. Antimicrobial Activity of Bitter Ginger and Galangal Extracts

The antimicrobial activities of bitter ginger and galangal extracts were investigated using acne-causing bacteria, including *C. acnes* DMST 14916, *S. aureus* TISTR 746, and *S. epidermis* TISTR 518, via a broth-microdilution assay. As in Table 2, most ethanol and methanol extracts of bitter ginger and galangal possessed bactericidal effects against *C. acnes*. The extracts of the plant rhizomes showed the broadest spectrum of antimicrobial activity against *C. acnes*, *S. aureus*, and *S. epidermidis*. The MICs of BRhE, BRhM, and GRhM against *C. acnes* were 3.96 mg/mL, while that of GRhE was 7.92 mg/mL. The MBCs of BRhE, BRhM, GRhE, and GRhM against *C. acnes* were 3.96 mg/mL, 7.92 mg/mL, 15.84 mg/mL, and 7.92 mg/mL, respectively. The MICs of BRhE, BRhM, GRhE, and GRhM against *S. aureus* were 7.92 mg/mL, 15.84 mg/mL, >31.68 mg/mL, and 31.68 mg/mL, while their MBCs were 7.92 mg/mL, >31.68 mg/mL, >31.68 mg/mL, and >31.68 mg/mL, respectively. The MICs of BRhE, BRhM, and GRhE against *S. epidermidis* were 15.84 mg/mL, while that of GRhM was 7.92 mg/mL. The MBCs of BRhE, BRhM, GRhE, and GRhM on *S. epidermidis* were >31.68 mg/mL. The extracts of the plant stems and leaves exhibited antimicrobial activity against *C. acnes* but did not affect *S. aureus* and *S. epidermidis*. The results indicated that the extracts of bitter ginger rhizome (BRhE and BRhM) and gingeral rhizome (GRhE, and GRhM) possess broad antimicrobial activity against these acne-causing bacteria. Therefore, these extracts were chosen for the next experiments in this study.

### 2.3. Effects of Bitter Ginger and Galangal Extracts Observed via SEM

The antimicrobial effects of the ethanol and methanol extracts of bitter ginger and galangal rhizomes were measured using *C. acnes* DMST 14916 and *S. epidermidis* TISTR 518 using scanning electron microscopy (SEM), and the results are shown in Figure 2 and Figure 3, respectively. As the results show in Figure 2, an untreated cell of *C. acnes* was used as a control cell, approximately 1 micron in size, and revealed a smooth surface without any rupture (Figure 2a). *C. acnes* cells treated with the rhizome extracts, such as BRhE, GRhE, BRhM, and GRhM, had obvious damage and shrinkage on the cell surfaces (arrows) (Figure 2b–e). As the results show in Figure 3, untreated cells of *S. epidermidis* were used as a control, which were round-shaped, approximately 1 micron in size, and had a smooth surface without any abnormality (Figure 3a). *S. epidermidis* cells treated with BRhE showed obvious shrinkage of the cell surface (arrows) (Figure 3b), while those treated with BRhM showed slight shrinkage and rupture (arrows) (Figure 3d). The cells treated with GRhE and GRhM revealed slight shrinkage but with much cell debris remaining on the cell surfaces (arrows) (Figure 3c,e).

### 2.4. Skin-Related Cytotoxicity of Bitter Ginger and Galangal Extracts

BRhE, BRhM, GRhE, and GRhM with the broad antimicrobial activity against acne-causing bacteria were taken for the cytotoxicity tests using HaCaT and MRC-5 cell lines, determined via an MTT assay. The results of the cytotoxicity against HaCaT cells are shown in Figure 4. The cytotoxicity of BRhE on HaCaT was present initially at 250 μg/mL but not significantly different from that of the untreated cells (*p >* 0.05) (Figure 4a). The cytotoxicity of GRhE and BRhM was not found between 62.5 μg/mL and 1000 μg/mL (Figure 4b,c). GRhM was not toxic to HaCaT cells until 500 μg/mL but slightly cytotoxic at 1000 μg/mL (*p* < 0.05) (Figure 4d). The results were correlated with the investigation of cell morphology after treatment with the extracts. The morphological appearances of HaCaT cells stained using the methylene blue technique were visible (Figure 5). HaCaT cells treated with BRhE (Figure 5b), GRhE (Figure 5c), BRhM (Figure 5d), and GRhM (Figure 5e) at 1000 μg/mL showed normal shapes similar to the untreated cells (Figure 5a). The cytotoxicity of bitter ginger and galangal extracts against MRC-5 cells is shown in Figure 6. BRhE, GRhE, BRhM, and GRhM were not toxic to MRC-5 cells until 1000 μg/mL (*p* > 0.05) (Figure 6a–d). The morphological appearances of MRC-5 cells were investigated as shown in Figure 7. MRC-5 cells treated with BRhE (Figure 7b), GRhE (Figure 7c), BRhM (Figure 7d), and GRhM (Figure 7e) at 1000 μg/mL showed normal shapes similar to the untreated cells (Figure 7a).

### 2.5. Phytochemicals in Bitter Ginger and Galangal Extracts Observed via GC-MS and LC-MS/MS

Volatile compounds in bitter ginger and galangal rhizome extracts identified via GC-MS are shown in Figure 8. (R)-lavandulyl (R)-2-methylbutanoate (RT: 27.24 min) and Zerumbone [2,6,10-cycloundecatrient-1-one, 2,6,9,9-tetramethyl-, (E,E,E)] (RT: 37.85 min) were found in both BRhE (Figure 8a) and BRhM (Figure 8c), while (s)-4-(1Acetoxyallyl)phenyl acetate (RT: 34.65 min) was found in both GRhE (Figure 8b) and GRhM (Figure 8d).

As with LC-MS chromatograms (Figure 9), phytochemical compounds obtained from LC-QTOF-MS/MS were analyzed in the bitter ginger rhizome extracts (Table 3 and Table 4). Results shown in Table 3 demonstrate that twenty phytochemicals in BRhE were identified, including sugar (allose and D-(+)-turanose), fatty acid derivatives (3-hydroxyphenyl-valeric acid), phenolic derivatives (1,3-dicaffeoylquinic acid and piceatannol 4′-galloylglucoside), flavonoid derivatives (apigenin 7-galactoside, 8-C-beta-D-glucofuranosylapigenin 2″-O-acetate, myricetin 3-(2″-p-hydroxybenzoylrhamnoside), tectorigenin and cirsimaritin), alkaloid (piperic acid), ubiquinones (myrsinone), catecholamine (n-acetyldopamine), 6a-hydroxymaackiain, canescacarpin, lauryl hydrogen sulfate, trifluoroacetic acid, N-undecylbenzene sulfonic acid, sodium tetradecyl sulfate, and 2-dodecylbenzene sulfonic acid. Similarly, Table 4 indicates twenty phytochemicals that were identified in BRhM, including sugar (sucrose), fatty acid derivatives (3-hydroxyphenyl-valeric acid), phenolic derivatives (1,3-dicaffeoylquinic acid, piceatannol 4′-galloylglucoside, 2,4,2′-trihydroxy-6″,6″-dimethyl-3′-prenylpyrano[2″,3′:4′,5′]chalcone), flavonoid derivatives (apigenin 7-galactoside, 8-C-beta-D-glucofuranosylapigenin 2″-O-acetate, myricetin 3-(2″-p-hydroxybenzoylrhamnoside), tectorigenin and cirsimaritin), ubiquinones (myrsinone), catecholamine (n-acetyldopamine), diterpenoids (triptophenolide), trifluoroacetic acid, N-undecylbenzene sulfonic acid, 2-dodecylbenzene sulfonic acid, cis-β-D-glucosyl-2-hydroxycinnamate, demethoxycurcumin, thyrotropin releasing hormone, and 7E,9E,11-dodecatrienyl acetate.

Phytochemicals in the galangal rhizome extracts are presented in Table 5 and Table 6. As the results show in Table 5, twenty phytochemicals in GRhE were identified and included sugar (sucrose), fatty acid derivatives (3-hydroxyphenyl-valeric acid), phenolic compounds and derivatives (sweroside and methylsyringin), flavonoids and flavonoid derivatives (amoritin and (+)-myristinin A), ubiquinones (myrsinone), catecholamine (n-acetyldopamine), diterpene and diterpenoids (sagequinone methide A and gamma-crocetin), coumarins (dihydrosamidin and phenprocoumon), sesquiterpenes (10-hydroxymelleolide), nivalenol, 2-(4-allyl-2,6-dimethoxyphenoxy)-1-(4-hydroxy-3-methoxyphenyl)-1-propanol, p-(3,4-dihydro-6-methoxy-2-naphthyl)phenol, (2-butylbenzofuran-3-yl) (4-hydroxyphenyl)ketone, thyrotropin releasing hormone, cortisone acetate, and dinoterb. As the results show in Table 6, twenty phytochemicals in BRhM were identified and included sugar (sucrose), phenolic compounds and derivatives (myzodendrone), flavonoids and flavonoid derivatives (neobavaisoflavone), ubiquinones (myrsinone), coumarins (phenprocoumon), diterpene and diterpenoids (sagequinone methide A), sesquiterpenes and derivatives (molephantinin), lignan (Gmelinol), 2-dodecylbenzenesulfonic acid, triptophenolide, lauryl hydrogen sulfate, 2-(4-Allyl-2,6-dimethoxyphenoxy)-1-(4-hydroxy-3-methoxyphenyl)-1-propanol, (2-Butylbenzofuran-3-yl)(4-hydroxyphenyl)ketone, gibberellin A120, nivalenol, thyrotropin releasing hormone, sodium tetradecyl sulfate, and N-undecylbenzenesulfonic acid.

## 3. Discussion

This study aimed to assess the potential for bitter ginger and galangal to be used to control acne vulgaris. Selection of the extraction solvent is a crucial factor affecting the efficiency of solid–liquid extraction techniques [24]. The percentage yields of ethanolic and methanolic extracts of galangal rhizomes in this study were lower than those reported by Boonkusol and coworkers, who achieved the yields of 17.66% and 16.85% via ethanol and methanol extraction, respectively, by soaking at room temperature for 24 h [25]. In this study, methanolic extracts of bitter ginger rhizomes and galangal stems had higher yields than their ethanolic extracts, while the ethanolic extract of bitter ginger possessed a yield greater than its methanolic extracts. However, most obtained yields of bitter ginger and galangal extracts were similar between the ethanol and methanol extraction.

From previous studies, the aqueous and ethanolic extracts of bitter ginger rhizomes extracted by soaking at 40 °C in an incubator shaker at 200 rpm for 5 days were used for antimicrobial activity against four multidrug resistant (MDR) bacteria (*Lactobacillus acidophilus*, *Streptococcus mutans*, *Enterococcus faecalis*, and *Staphylococcus aureus*) through disc diffusion. The aqueous and ethanolic extracts showed synergy with antibiotics indicating the potential to combine topical extracts with systemic antibiotics for the treatment of acne [26]. Similarly, aqueous and ethanolic extracts of bitter ginger rhizomes obtained using the water bathing technique possessed antimicrobial activity against *S. mutans*, *E. faecalis*, *Staphylococcus* spp., and *Lactobacillus* spp., based on a disc diffusion technique [27]. This indicates that the observed antimicrobial activity of bitter ginger rhizomes is likely attributable to a cocktail of plant-derived compounds.

Here, bitter ginger rhizomes were extracted with a 70% ethanol solvent using the maceration method for 24 h. The bitter ginger extracts at concentrations, including 5%, 10%, and 15%, showed antibacterial activity against *C. acnes* (*P. acnes*) at 5.53%, 7.30%, and 8.07%, respectively, based on a disc diffusion assay [28]. Galangal rhizomes were dried and extracted with ethyl acetate and methanol under reflux conditions for 1 h (×2). They were assessed for antimicrobial activity against acne-causing bacteria. The MICs of ethyl acetate and methanolic extracts of galangal rhizomes against *C. acnes* (*P. acnes*), *S. aureus*, and *S. epidermidis* were 156.0 and >5.0 × 10^3^ μg/mL, 625.0 and >5.0 × 10^3^ μg/mL, and 625.0 and >5.0 × 10^3^ μg/mL, respectively [29]. Significantly, in our study we reported that the extracts of the different parts used of bitter ginger and galangal, such as rhizomes, stems, and leaves, could have antimicrobial potential against *C. acnes*, as shown in Table 2. Previously this has not been assessed and demonstrates the usefulness of the entire plant for providing a sustainable antimicrobial.

Galangal rhizomes were also extracted with an ethanol solvent by soaking at room temperature, overnight. The ethanolic extract had antibacterial activity against *S. aureus* 209P based on an agar disc diffusion method, and to understand the possible mechanism of activity, the physiological effects were observed via transmission electron microscopy (TEM). Cells of *S. aureus* treated with the galangal extract revealed some alterations to the cell membrane and some damage to the bacterial cell wall [30]. Hexane extracts of galangal rhizomes possessed antibacterial activity against *S. aureus* SA113. The galangal extracts possessed antibiofilm efficacy by reducing biofilm adherence, observed via SEM [31]. In our study, the results showed that the bitter ginger and galangal rhizome extracts could cause obvious shrinkage and rupture on cell surfaces of *C. acnes* and *S. epidermidis*, which were captured via SEM. This suggests that growth inhibition and/or killing is the result cell integrity disruption. The chemical constituents revealed via MS analysis included compounds that are known to be toxic or induce oxidative damage to biological materials, such as lipids, proteins, and DNA, which likely mediates the physiological changes we observed. For a topical antimicrobial to be useful, it must be efficacious against bacteria without damaging the host. In this study, the results indicated that the bitter ginger and galangal extracts could be safe for topical application, showing negligible toxicity against human keratinocytes and fibroblasts. This is in keeping with the current literature, which indicates moderate toxicity, dependent on the solvent used for extraction.

Acne can lead to localized wounding of the skin. An antioxidant environment is key to promoting the wound-healing process by controlling local oxidative stress. Extracts of galangal and bitter ginger were demonstrably anti-oxidant, with some variation associated with the different solvent extraction methods. Several studies have demonstrated anti-oxidant activity, which has been attributed to specific plant-derived compounds. The extent of this activity is invariably dependent on the source, environmental conditions, processing, and extraction methods. A comprehensive analysis of the composition of bitter ginger and galangal extracts is warranted to establish key compounds that mediate this activity.

In summary, we present an analysis of bitter ginger and galangal extracts from the perspective of a skin topical to treat acne. The combined antimicrobial activity, anti-oxidant activity, and negligible toxicity suggest that these extracts could have a place in the management of acne. For applications, several obstacles remain to ensure that plant extracts have consistent activity, which is key to their successful implementation clinically. However, we demonstrate a proof of principle that bitter ginger and galangal could have a place in acne management in the future.

Furthermore, based on our mass spectrometric analyses (GC-MS and LC-Ms/MS), phytochemicals in the bitter ginger rhizome extracts, such as zerumbone [32,33], tectorigenin [34,35], piperic acid [36], cirsimaritin [37,38], demethoxycurcumin [39,40], and 1,3-dicaffeolquinic acid [41], while those in the galangal rhizome extracts, such as sweroside [42] and neobavaisoflavone [43,44], were expected to provide antioxidant and/or antimicrobial activities. The chemical structures of the phytochemicals are shown in Figure 10. Although the extracts of bitter ginger and galangal rhizomes could be suited for the development of topical anti-acne formulations, a stronger emphasis on the specific mechanisms through which the phytochemical compounds act on acne-causing bacteria is needed for further studies.

## 4. Materials and Methods

### 4.1. Acne-Causing Bacteria and Plant Materials

Bitter ginger (*Zingiber zerumbet* (L.) Roscoe) and galangal (*Alpinia galanga* (L.) Willd) were purchased from local agricultural farms from Nakhon Si Thammarat and Chiang Rai, Thailand, respectively during October–November 2021. The three acne-causing bacteria used were *Cutibacterium acnes* DMST 14916, *Staphylococcus epidermis* TISTR 518, and *Staphylococcus aureus* TISTR 746. *C. acnes* was cultured in brain heart infusion broth under an anaerobic condition at 37 °C for 3–5 days. *S. aureus* and *S. epidermidis* were cultured in nutrient broth at 37 °C for 24–48 h. The bacteria were obtained from the Biology and Biotechnology Laboratory of the Scientific and Technological Instruments Center, Mae Fah Luang University, Thailand.

### 4.2. Plant Preparation and Extraction

The rhizomes, stems, and leaves of bitter ginger and galangal were separated, washed with tap water, and cut into small pieces. Small pieces of each part were dried using a tray dryer at 60 °C until complete dryness. The dried plant pieces were ground into powder using a hammer mill. The extraction method of bitter ginger and galangal was modified slightly from the previous study [45]. Bitter ginger and galangal powder samples (30 g) were taken separately for extraction using absolute ethanol or methanol at 1:6 (*w*/*v*). The extraction samples were incubated in an incubator shaker at room temperature, at 150 rpm for 24 h. Then, the mixture was filtered through Whatman^®^ No.1 filter papers (Cytiva, Shanghai, China). The filtrate samples were taken to a rotary evaporator at 60 °C to remove the extraction solvents. The crude extracts were kept until use. The yields of crude extracts were calculated with triplication.

### 4.3. Antioxidant Activity Assay

The antioxidant activity of bitter ginger and galangal extracts was evaluated by performing a DPPH assay with a slight modification from the previous study [45]. Briefly, various concentrations of bitter ginger and galangal extracts (50 μL) were added to 200 μL of a 0.1 mM DPPH solution. The reactions were incubated under dark conditions at room temperature for 30 min. The absorbance was measured at 517 nm using a microplate reader. Ascorbic acid was used as a positive control. The percent inhibition of antioxidants activity (I%) was calculated by using the equation of I% = [(A_517_ control − A_517_ sample)/A_517_ control] × 100. Where, A_517_ control is the absorbance of the control solution without any sample and A_517_ sample is the absorbance of the solution with bitter ginger and galangal extracts (or ascorbic acid). The IC_50_ value is the concentration of bitter ginger and galangal extracts (or ascorbic acid) required to inhibit antioxidant activity by 50%.

### 4.4. Antimicrobial Activity Assay

The antimicrobial activity of bitter ginger and galangal extracts against *C. acnes* DMST 14916, *S. epidermidis* TISTR 518, and *S. aureus* TISTR 746 was tested using the broth micro-dilution assay, which was slightly modified from previous studies [46,47]. Briefly, the concentrations of bitter ginger and galangal extracts with serial dilutions were prepared in 10% DMSO. Bacterial cells were cultured to log phase (OD_600nm_ = 0.5–0.8) and diluted to the density at approximately 10^6^ cells/mL (OD_600nm_ = 0.001). The microbial cells were treated with the various concentrations of bitter ginger and galangal extracts (0.50–31.68 mg/mL) and then incubated at 37 °C for 24 h for *S. epidermidis* and *S. aureus* and 72 h for the anaerobic condition for *C. acnes*. DMSO (10% *v*/*v*) was used as a negative control, while tetracycline was used as a positive control.

The minimum inhibitory concentration (MIC) values of bitter ginger and galangal extracts were measured using a resazurin dye solution technique [48]. After 24 h or 72 h incubation, the 0.06% resazurin dye solution (10 μL) was added to the bacterial tests and incubated under the same conditions for 4–6 h. The MIC value was the lowest concentration of plant extracts that could inhibit microbial growth without changing the coloration of the resazurin dye. The maximum bactericidal concentration (MBC) values of the plant extracts were further evaluated using the bacteria by performing a colony plate count technique.

### 4.5. Cytotoxic Activity Assay

The cytotoxic activities of bitter ginger and galangal extracts, including BRhE, GRhE, BRhM, and GRhM, were investigated using human cell lines via an MTT assay, which was slightly modified from the previous study [49]. Human keratinocyte HaCaT and fibroblast MRC-5 cells (approximately 1 × 10^4^ cells/well) were seeded onto 96-well plates in RPMI-1640 medium and incubated at 37 °C under a humidified condition of 5% CO_2_ for 24 h. Cell viability was tested in the presence of plant extract concentrations (62.5–1000 μg/mL) and incubated at the same condition for 24 h. The tests were incubated with 150 μL of 0.5 mg/mL MTT solution for 1 h at 37 °C in a humidified condition to cause a purple-colored formazan salt product. The DMSO solution (100 μL) was mixed into the tests to solubilize the formazan salt product. The solubilized samples of each test were taken for measurements at 550 nm. The % cell viability was calculated by comparing the absorbance values of plant extract-treated and untreated cells. The untreated cells were used as an experimental control. All experiments were repeated at least three times.

### 4.6. Methylene Blue Staining

The morphological evaluation of bitter ginger and galangal rhizome extracts based on HaCaT and MRC-5 cells was carried out using the methylene blue staining method [50]. The cells treated with the extracts were washed with ice-cold PBS, fixed with 50% (*v*/*v*) ice-cold ethanol solution, and stained with 0.2% (*w*/*v*) methylene blue solution for 30 s. The solution was aspirated after that, and the cells were washed with ice-cold water three times. The cell samples were dried and observed under a light microscope.

### 4.7. Scanning Electron Microscopic Analysis

The antimicrobial effects of bitter ginger and galangal extracts on *C. acnes* DMST 14916 and *S. epidermidis* TISTR 518 were investigated via scanning electron microscopy (SEM) with a modified method of the previous study [51]. Briefly, the microbial cells were grown to log phase (OD_600nm_ = 0.5–0.8), centrifuged at 3500× *g* for 3 min, and washed twice with PBS, pH 7.4. The cells were harvested and diluted to the density of approximately 10^8^ cells/mL (OD_600nm_ = 0.1). The diluted *C. acnes* and *S. epidermidis* were incubated in the presence of bitter ginger and galangal extracts at 10× MICs for 60 min under anaerobic and aerobic conditions, respectively. Cells without any treatment were controls. Each sample test (10 μL) was smeared on a cover slide and then fixed by moving it through a flame. The bacterial cells were dried gradually by adding a series of ethanol solutions, including 30%, 50%, 70%, 90%, 90%, 100%, and 100%, respectively, for 30–60 min in each solution. The dried bacterial cells were coated with gold-palladium and captured under a Field Emission Scanning Electron microscope (TESCAN MIRA4, Brno, Czech Republic).

### 4.8. GC-MS Analysis

Volatile compounds in bitter ginger and galangal extracts were analyzed using the GC-MS method, which was slightly modified from the previous study [52]. The GC samples of plant extracts (500 ppm) were prepared in absolute methanol, filtered through a 0.2 μm Econofilter, and filled into 1.5 mL glass vials. The samples were injected into the GC column (Agilent 6890N HP-5MS, Santa Clara, CA, USA, 0.25 mm × 30 mm × 0.25 μm). The oven temperature was set at an initial temperature of 60 °C and a temperature up to 325 °C. Helium was a carrier gas with a flow rate of 1.0 mL/min. The Agilent 6890N MS operation was performed to compute the retention time (RT) and corrected peak areas in each spectrum. Compounds were identified by matching the retention time (RT) of eluted peaks on the GC column with mass spectra via a comparison with NIST and WILEY library databases.

### 4.9. LC-MS/MS Analysis

The phytochemicals in bitter ginger and galangal extracts were analyzed using an LC-MS/MS method, which was used as the previous study [53]. Bitter ginger and galangal extract samples (500 ppm) were prepared in absolute methanol, filtered through 0.2 μm NYL filters, and collected into 1.5 mL glass vials. For LC operating conditions, the extract samples were injected into an Agilent Poroshell EC-C18 column (2.1 mm × 150 mm, 2.7 µm) with an Agilent Poroshell EC-C18 guard column (4.6 mm × 5 mm, 2.7 µm), operated by the Agilent 1290 UHPLC system (Agilent Technologies, Santa Clara, CA, USA). The LC separation was performed under a time and gradient program in which mobile phases were composed of 0.1% (*v*/*v*) formic acid in water (mobile phase A) and in acetonitrile (mobile phase B) at a flow rate of 0.2 mL/min. For MS acquisition, the data were obtained with an Agilent G6454B Q-TOF Mass Spectrometry unit (Agilent Technologies, Santa Clara, CA, USA) containing a Dual AJS ESI ion source, 4000 V of capillary voltage (VCap), and 500 V of nozzle voltage. The voltages of the skimmer1, fragmentor, and OctopoleRFPeak were set at 65 V, 150 V, and 750 V, respectively. The scan range was 100–1100 *m*/*z*. The scan rate was 1.00 spectra/s. The internal reference compounds with *m*/*z* 121.05087300 and *m*/*z* 922.00979800 for the positive mode and m/z 112.98558700 and *m*/*z* 1033.98810900 for the negative mode were used as Agilent reference masses. For MS/MS acquisition, the data were obtained by setting at the same parameters of the MS acquisition and at 10, 20, or 40 eV of collision energy.

### 4.10. Statistical Analysis

The differences between control and sample groups were measured using the statistical software Statistix ver. 9.0. Comparisons among groups were performed based on an analysis of variance using the ANOVA test. Significant difference analysis between the control and sample groups was performed via the student’s *t*-test, and the significance was considered when the *p-*value was less than 0.05 (* *p* < 0.05).

## 5. Conclusions

The current study has investigated the in vitro skin-related cytotoxic, antioxidant, and antimicrobial activities of bitter ginger and galangal extracts against acne-causing bacteria, in addition to identifying the activity-related phytochemicals via GC-MS and LC-MS/MS. The extracts of bitter ginger and galangal’s rhizomes, stems, and leaves possessed DPPH radical scavenging ability, whereas only the rhizome extracts of both plants were broadly antimicrobial against *C. acnes* DMST 14916, *S. aureus* TISTR 746, and *S. epidermis* TISTR 518. Through an SEM observation, the rhizome extracts revealed rupturing and shrinking effects on the cell surfaces of *C. acnes* and *S. epidermidis*. The extracts were also found to be non-toxic or slightly toxic to human keratinocytes (HaCaT) and fibroblasts (MRC-5) at the high concentration. Phytochemicals in the bitter ginger rhizome extracts, such as zerumbone (GC-MS), tectorigenin, piperic acid, cirsimaritin, demethoxycurcumin, and 1,3-dicaffeolquinic acid (LC-MS/MS), while those in the galangal rhizome extracts, such as sweroside and neobavaisoflavone (LC-MS/MS), were expected to provide antioxidant and antimicrobial activities. This investigation demonstrated bitter ginger and galangal extracts as having a high potential source for active antioxidant ingredients to neutralize free radicals, as well as natural antimicrobial compounds to treat acne-causing bacteria.

## Figures and Tables

**Figure 1 ijms-25-10869-f001:**
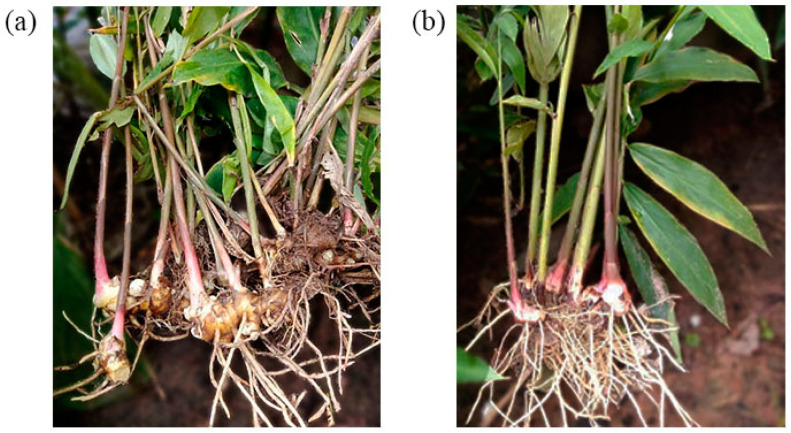
Whole plants of bitter ginger (*Zingiber zerumbet* (L.) Roscoe) (**a**) and galangal ((*Alpinia galanga* (L.) Willd) (**b**).

**Figure 2 ijms-25-10869-f002:**
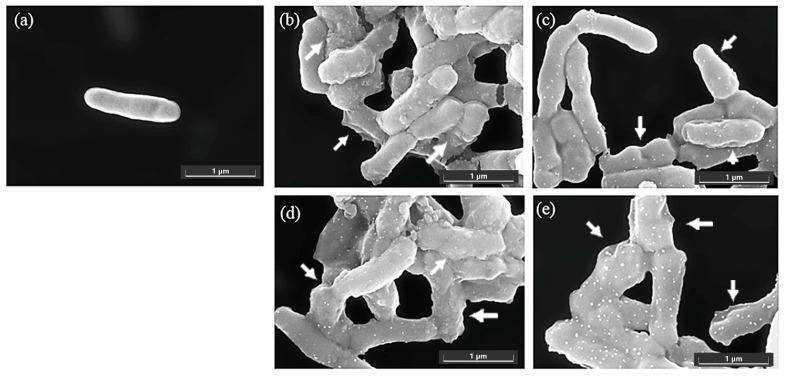
SEM images of *Cutibacterium acnes* DMST 14916 after incubation with bitter ginger and galangal extracts at 10× MICs for 60 min. The density of bacterial cells was used at approximately 1 × 10^8^ cells/mL. Cells of *C. acnes* were treated without any sample as controls (**a**). The cells were treated with the ethanol extracts of the bitter ginger rhizome (BRhE) (**b**) and galangal rhizome (GRhE) (**c**) and the methanol extracts of the bitter ginger rhizome (BRhM) (**d**) and galangal rhizome (GRhM) (**e**). Scale bar: 1 μm.

**Figure 3 ijms-25-10869-f003:**
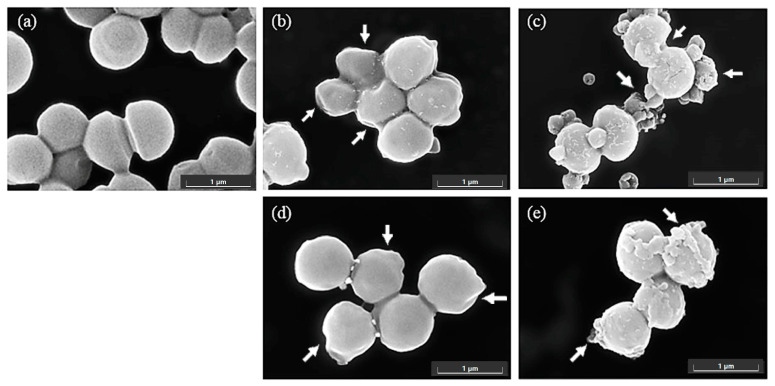
SEM images of *Staphylococcus epidermidis* TISTR 518 after incubation with bitter ginger and galangal extracts at 10× MICs for 60 min. The density of bacterial cells was used at approximately 1 × 10^8^ cells/mL. Cells of *S. epidermidis* were treated without any sample as controls (**a**). The cells were treated with the ethanol extracts of the bitter ginger rhizome (BRhE) (**b**) and galangal rhizome (GRhE) (**c**) and the methanol extracts of the bitter ginger rhizome (BRhM) (**d**) and galangal rhizome (GRhM) (**e**). Scale bar: 1 μm.

**Figure 4 ijms-25-10869-f004:**
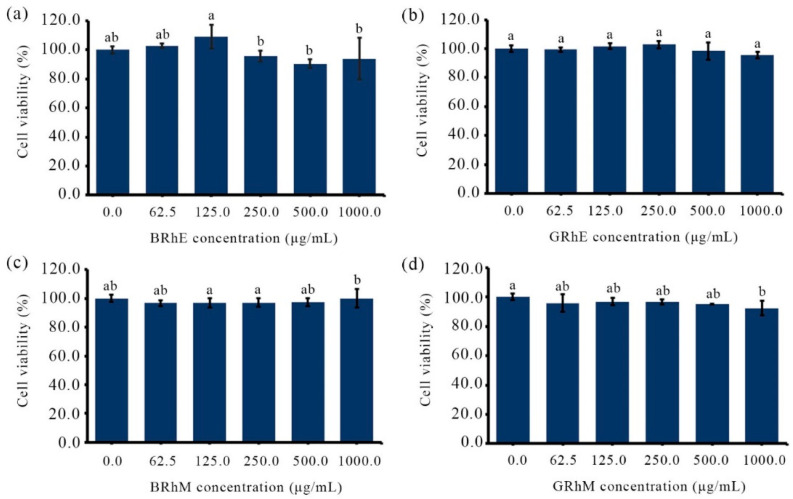
Cytotoxicity activities of bitter ginger and galangal extracts against HaCaT determined via an MTT assay. The cell density of HaCaT was used at approximately 1 × 10^4^ cells/well. The cells were treated with the ethanol extracts of the bitter ginger rhizome (BRhE) (**a**) and galangal rhizome (GRhE) (**b**) and the methanol extracts of the bitter ginger rhizome (BRhM) (**c**) and galangal rhizome (GRhM) (**d**). The superscript letters indicate significant (*p* < 0.05) differences of means within the plant extracts.

**Figure 5 ijms-25-10869-f005:**
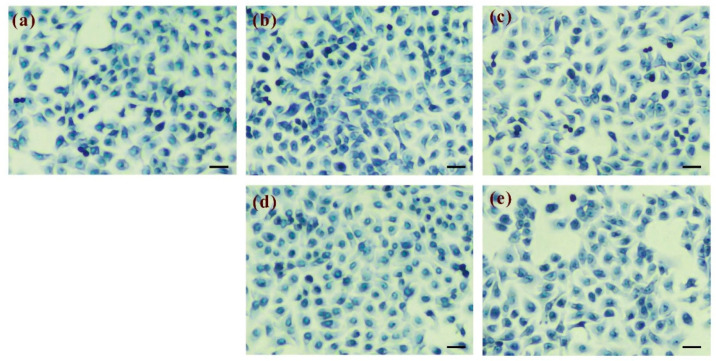
Microscopic examination of morphology of HaCaT cells after treatment with the bitter ginger and galangal rhizome extracts obtained using the methylene blue staining technique. HaCaT cells without any treatment (untreated cells) (**a**). The cells were treated with the ethanol extracts of the bitter ginger rhizome (BRhE) (**b**) and galangal rhizome (GRhE) (**c**) and the methanol extracts of the bitter ginger rhizome (BRhM) (**d**) and galangal rhizome (GRhM) (**e**) at the highest test concentration of 1000 μg/mL. Scale bar: 20 μm.

**Figure 6 ijms-25-10869-f006:**
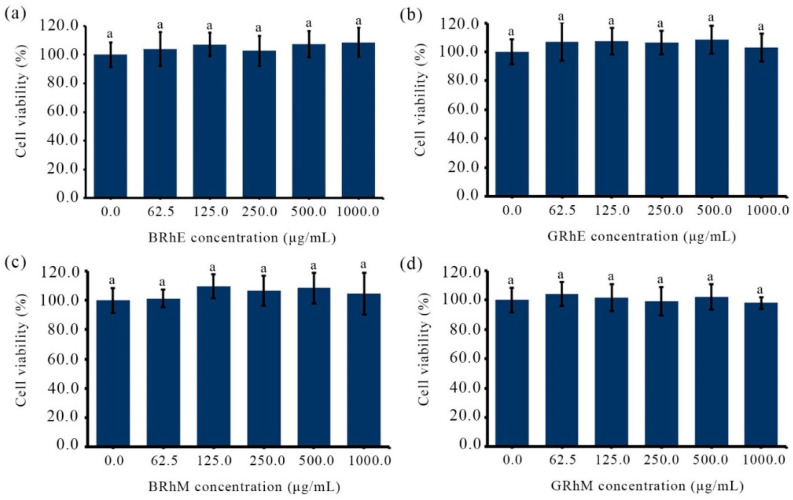
Cytotoxicity activities of bitter ginger and galangal extracts against MRC-5 determined via an MTT assay. The cell density of MRC-5 was used at approximately 1 × 10^4^ cells/well. The cells were treated with the ethanol extracts of the bitter ginger rhizome (BRhE) (**a**) and galangal rhizome (GRhE) (**b**) and the methanol extracts of the bitter ginger rhizome (BRhM) (**c**) and galangal rhizome (GRhM) (**d**). The superscript letter indicates significant (*p* < 0.05) differences of means within the plant extracts.

**Figure 7 ijms-25-10869-f007:**
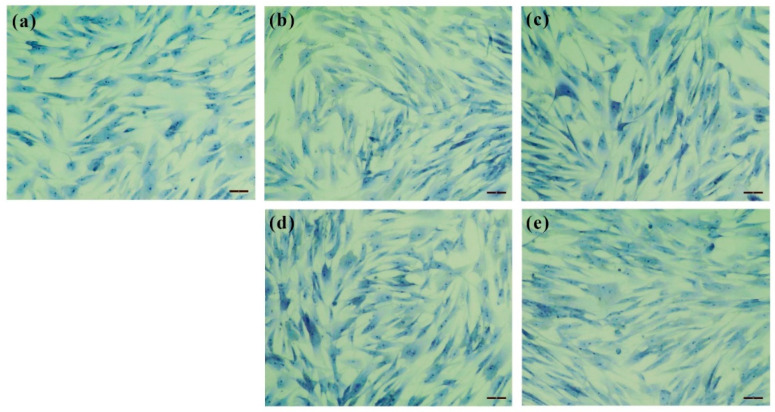
Microscopic examination of morphology of MRC-5 cells after treatment with the bitter ginger and galangal rhizome extracts obtained via the methylene blue staining technique. MRC-5 cells without any treatment (**a**). The cells were treated with the ethanol extracts of the bitter ginger rhizome (BRhE) (**b**) and galangal rhizome (GRhE) (**c**) and the methanol extracts of the bitter ginger rhizome (BRhM) (**d**) and galangal rhizome (GRhM) (**e**) at the highest test concentration of 1000 μg/mL. Scale bar: 100 μm.

**Figure 8 ijms-25-10869-f008:**
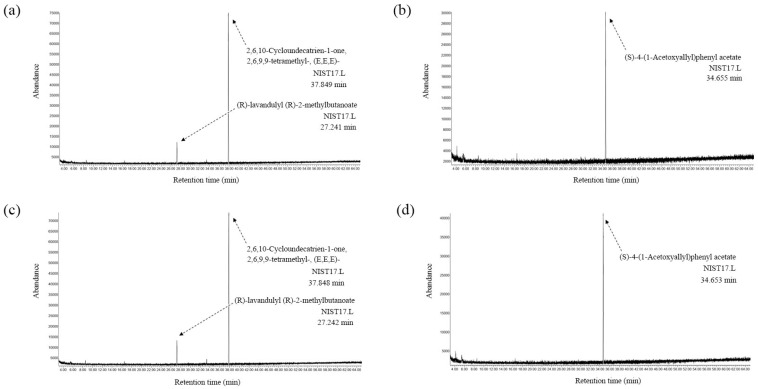
Volatile compounds in bitter ginger and galangal extracts determined via GC-MS. The GC chromatograms of the ethanol extracts of the bitter ginger rhizome (BRhE) (**a**) and galangal rhizome (GRhE) (**b**) and the methanol extracts of the bitter ginger rhizome (BRhM) (**c**) and galangal rhizome (GRhM) (**d**).

**Figure 9 ijms-25-10869-f009:**
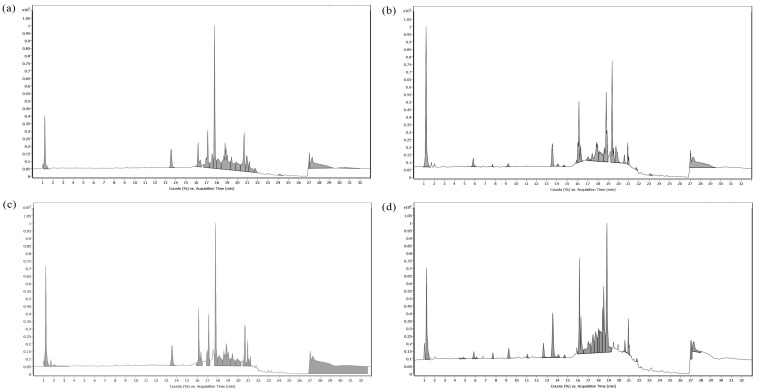
LC-MS chromatograms of ethanolic and methanolic extracts of bitter ginger and galangal rhizomes. The LC-MS chromatograms of the ethanol extracts of the bitter ginger rhizome (BRhE) (**a**) and galangal rhizome (GRhE) (**b**) and the methanol extracts of the bitter ginger rhizome (BRhM) (**c**) and galangal rhizome (GRhM) (**d**).

**Figure 10 ijms-25-10869-f010:**
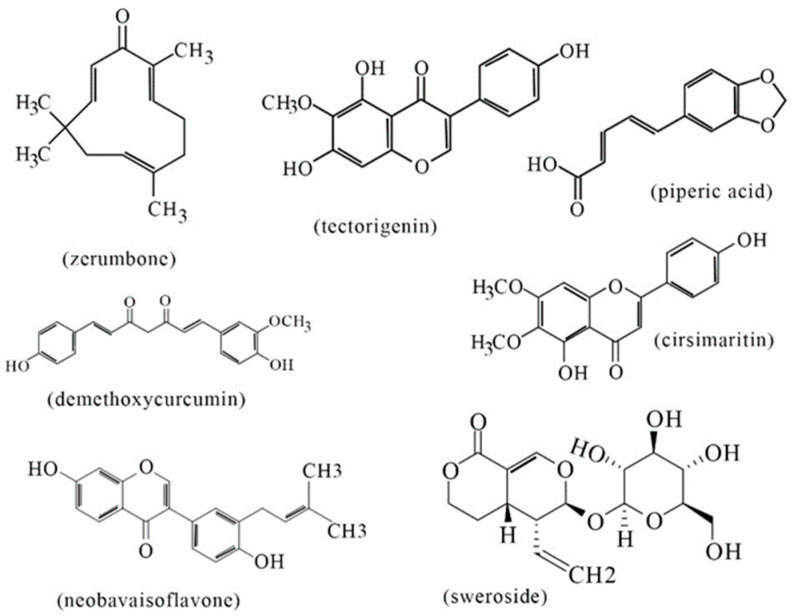
Major phytochemical constituents with antioxidant and/or antimicrobial activities in the bitter ginger and galangal rhizome extracts.

**Table 1 ijms-25-10869-t001:** Yields and DPPH radical scavenging activity of crude extracts of galangal and bitter ginger.

Extracts	Yields ± SD (%)	IC_50_ ± SD (mg/mL)
Ethanolic extraction		
Bitter ginger rhizome (BRhE)	5.71 ± 0.63 ^b^	1.19 ± 0.06 ^b^
Bitter ginger stem (BStE)	5.03 ± 0.76 ^c^	1.42 ± 0.04 ^a^
Bitter ginger leaf (BLeE)	2.14 ± 0.34 ^de^	0.40 ± 0.02 ^e^
Galangal rhizome (GRhE)	5.47 ± 0.06 ^b^	0.08 ± 0.01 ^h^
Galangal stem (GStE)	1.72 ± 0.06 ^e^	0.15 ± 0.01 ^g^
Galangal leaf (GLeE)	5.37 ± 0.94 ^b^	0.27 ± 0.03 ^f^
Methanolic extraction		
Bitter ginger rhizome (BRhM)	7.30 ± 0.09 ^a^	0.99 ± 0.04 ^c^
Bitter ginger stem (BStM)	1.06 ± 0.13 ^f^	0.46 ± 0.03 ^d^
Bitter ginger leaf (BLeM)	2.14 ± 0.37 ^de^	0.30 ± 0.01 ^f^
Galangal rhizome (GRhM)	6.94 ± 0.50 ^b^	0.06 ± 0.01 ^h^
Galangal stem (GStM)	2.72 ± 0.62 ^d^	0.28 ± 0.02 ^f^
Galangal leaf (GLeE)	5.67 ± 0.36 ^b^	0.17 ± 0.02 ^g^
Ascorbic acid	-	1.4 ± 0.2 *

* The value is expressed in μg/mL. Values (mean ± SD) are the averages of three samples of each plant extract, analyzed individually in triplicate. Superscript letters within the same column indicate significant (*p* < 0.05) differences of means within the plant extracts.

**Table 2 ijms-25-10869-t002:** The antimicrobial activity of crude extracts of galangal and bitter ginger.

Extracts	*Cutibacterium acnes* DMST 14916		*Staphylococcus aureus* TISTR 746		*Staphylococcus epidermis* TISTR 518	
	MIC (mg/mL)	MBC (mg/mL)	MIC (mg/mL)	MBC (mg/mL)	MIC (mg/mL)	MBC (mg/mL)
Ethanol extraction						
Bitter ginger rhizome (BRhE)	3.96	3.96	7.92	7.92	15.84	>31.68
Bitter ginger stem (BStE)	31.68	>31.68	Nd	Nd	Nd	Nd
Bitter ginger leaf (BLeE)	31.68	31.68	Nd	Nd	Nd	Nd
Galangal rhizome (GRhE)	7.92	15.84	>31.68	>31.68	15.84	>31.68
Galangal stem (GStE)	>31.68	>31.68	Nd	Nd	Nd	Nd
Galangal leaf (GLeE)	7.92	31.68	Nd	Nd	Nd	Nd
Methanol extraction						
Bitter ginger rhizome (BRhM)	3.96	7.92	15.84	>31.68	15.84	>31.68
Bitter ginger stem (BStM)	>31.68	>31.68	Nd	Nd	Nd	Nd
Bitter ginger leaf (BLeM)	3.96	>3.96	Nd	Nd	Nd	Nd
Galangal rhizome (GRhM)	3.96	7.92	31.68	>31.68	7.92	>31.68
Galangal stem (GStM)	15.84	15.84	Nd	Nd	Nd	Nd
Galangal leaf (GLeE)	15.84	15.84	Nd	Nd	Nd	Nd
Tetracycline	1 *	4 *	2 *	4 *	64 *	256 *

Nd, not detected in concentration range of 0.50–31.68 mg/mL. * The value is expressed in μg/mL.

**Table 3 ijms-25-10869-t003:** Analysis of phytochemical constituents in the ethanol extract of bitter ginger rhizome (BRhE) performed via LC-QTOF-MS-MS.

RT (min)	*m*/*z*	MS/MS Fragments	Formula	Tentative Identification	Mass	Ion Species
1.201	179.0566	59.0144, 71.0144	C6 H12 O6	Allose	180.0640	(M-H)-
1.264	341.1094	89.0249, 179.0552, 341.1089	C12 H22 O11	D-(+)-Turanose	342.1166	(M-H)-
16.393	431.0987	285.0402, 431.0984	C21 H20 O10	Apigenin 7-galactoside	432.1059	(M-H)-
17.104	473.109	284.0323, 413.087, 473.107	C23 H22 O11	8-C-beta-D-Glucofuranosylapigenin 2″-O-acetate	474.1163	(M-H)-
17.835	515.1211	284.0326, 455.0974, 515.1196	C25 H24 O12	1,3-Dicaffeoylquinic acid	516.1281	(M-H)-
17.839	583.1079	284.0316, 515.1195, 583.1062	C28 H24 O14	Myricetin 3-(2″-p-hydroxybenzoylrhamnoside)	584.1149	(M-H)-
17.911	299.0563	112.9856, 284.0333, 300.0592	C16 H12 O6	Tectorigenin	300.0636	(M-H)-
18.512	193.0871	124.0155, 193.0870	C11 H14 O3	3-Hydroxyphenyl-valeric acid	194.0943	(M-H)-
18.589	557.1303	284.0324, 557.1307	C27 H26 O13	Piceatannol 4′-galloylglucoside	558.1375	(M-H)-
18.847	217.0508	68.9983, 158.0374, 173.0603	C12 H10 O4	Piperic acid	218.0581	(M-H)-
18.856	299.0559	63.0237, 151.0025, 255.0304	C16 H12 O6	6a-Hydroxymaackiain	300.0632	(M-H)-
18.919	293.1761	71.0141, 177.0915, 236.1057	C17 H26 O4	Myrsinone	294.1834	(M-H)-
18.929	337.1085	119.0503, 217.0506	C20 H18 O5	Canescacarpin	338.116	(M-H)-
19.104	313.072	112.9856, 283.0243	C17 H14 O6	Cirsimaritin	314.0792	(M-H)-
19.512	194.0823	180.0603, 194.0822	C10 H13 N O3	n-acetyldopamine	195.0896	(M-H)-
19.949	265.1482	96.9603, 265.1479	C12 H26 O4 S	Lauryl hydrogen sulfate	266.1555	(M-H)-
20.322	112.9856	68.9961	C2 H F3 O2	trifluoroacetic acid	113.9929	(M-H)-
20.908	311.1691	183.0123, 311.1691	C17 H28 O3 S	N-Undecylbenzene sulfonic acid	312.1764	(M-H)-
21.790	293.1797	96.9605, 293.1794	C14 H30 O4 S	Sodium tetradecyl sulfate	294.1869	(M-H)-
21.827	325.1844	119.0504, 183.0124	C18 H30 O3 S	2-Dodecylbenzene sulfonic acid	326.1917	(M-H)-

**Table 4 ijms-25-10869-t004:** Analysis of phytochemical constituents of the methanol extract of bitter ginger rhizome (BRhM) performed via LC-QTOF-MS-MS.

RT (min)	*m*/*z*	MS/MS Fragments	Formula	Tentative Identification	Mass	Ion Species
1.254	341.1092	89.0243, 179.0555, 341.1091	C12 H22 O11	Sucrose	342.1165	(M-H)-
8.134	325.093	145.0294, 265.0748	C15 H18 O8	cis-β-D-Glucosyl-2-hydroxycinnamate	326.1003	(M-H)-
16.394	431.0992	255.0254, 285.0401, 431.0965	C21 H20 O10	Apigenin 7-galactoside	432.1063	(M-H)-
17.103	473.1095	284.0325, 413.0876, 473.1086	C23 H22 O11	8-C-beta-D-Glucofuranosylapigenin 2″-O- acetate	474.1166	(M-H)-
17.782	515.1215	284.0328, 455.0978, 515.1204	C25 H24 O12	1,3-Dicaffeoylquinic acid	516.1285	(M-H)-
17.828	583.1081	515.1198, 583.1049	C28 H24 O14	Myricetin 3-(2″-p-hydroxybenzoylrhamnoside)	584.1151	(M-H)-
17.834	299.0563	112.9853, 284.0324	C16 H12 O6	Tectorigenin	300.0636	(M-H)-
18.542	193.0869	53.0034, 177.0556	C11 H14 O3	3-Hydroxyphenyl-valeric acid	194.0942	(M-H)-
18.635	557.1306	284.0327, 497.1038, 557.1301	C27 H26 O13	Piceatannol 4′-galloylglucoside	558.1377	(M-H)-
18.828	337.1084	119.0505, 217.0507, 337.1076	C20 H18 O5	Demethoxycurcumin	338.116	(M-H)-
18.912	361.1635	71.0143, 236.1053	C16 H22 N6 O4	Thyrotropin releasing hormone	362.1708	(M-H)-
18.932	293.1762	71.0144, 236.1054, 293.1754	C17 H26 O4	Myrsinone	294.1835	(M-H)-
19.037	313.0720	255.0296, 283.0249, 313.0704	C17 H14 O6	Cirsimaritin	314.0793	(M-H)-
19.516	194.0825	61.9868, 135.0073, 194.0825	C10 H13 N O3	n-Acetyldopamine	195.0898	(M-H)-
20.11	405.1709	119.0503, 285.1133, 405.1706	C25 H26 O5	2,4,2′-Trihydroxy-6″,6″-dimethyl-3′-prenylpyrano[2″,3″:4′,5′]chalcone	406.1781	(M-H)-
20.442	221.1547	205.1226, 221.1543	C14 H22 O2	7E,9E,11-Dodecatrienyl acetate	222.162	(M-H)-
20.655	311.169	183.0122, 311.1685	C20 H24 O3	Triptophenolide	312.1759	(M-H)-
20.911	311.1689	183.0122, 311.1687	C17 H28 O3 S	N-Undecylbenzenesulfonic acid	312.1761	(M-H)-
21.14	112.9856	68.9962	C2 H F3 O2	Trifluoroacetic acid	113.9929	(M-H)-
21.889	325.1845	119.0508, 183.0128	C18 H30 O3 S	2-Dodecylbenzenesulfonic acid	326.1918	(M-H)-

**Table 5 ijms-25-10869-t005:** Analysis of phytochemical constituents of the ethanol extract of galangal rhizome (GRhE) performed via LC-QTOF-MS-MS.

RT (min)	*m*/*z*	MS/MS Fragments	Formula	Tentative Identification	Mass	Ion Species
1.261	341.1093	59.0141, 89.0245, 179.0559	C12 H22 O11	Sucrose	342.1166	(M-H)-
5.826	357.1194	149.0605, 357.1176	C16 H22 O9	Sweroside	358.1266	(M-H)-
16.007	311.1138	149.0606, 311.1186	C15 H20 O7	Nivalenol	312.121	(M-H)-
16.100	385.1505	101.0243, 177.0918, 385.1467	C18 H26 O9	Methylsyringin	386.1578	(M-H)-
17.468	387.1453	149.0607, 263.1074, 341.1392	C21 H24 O7	Dihydrosamidin	388.1525	(M-H)-
17.863	373.1661	251.1079, 327.1600	C21 H26 O6	2-(4-Allyl-2,6-dimethoxyphenoxy)-1-(4-hydroxy-3-methoxyphenyl)-1-propanol	374.1733	(M-H)-
17.868	251.1081	93.0342, 251.1070	C17 H16 O2	p-(3,4-Dihydro-6-methoxy-2-naphthyl)phenol	252.1152	(M-H)-
17.874	327.1605	251.1076, 327.1587	C20 H24 O4	Sagequinone methide A	328.1677	(M-H)-
18.068	279.1029	173.0607, 119.0499, 279.1021	C18 H16 O3	Phenprocoumon	280.1101	(M-H)-
18.178	293.1183	83.0498, 119.0503, 187.0762, 293.1170	C19 H18 O3	(2-Butylbenzofuran-3-yl) (4-hydroxyphenyl)ketone	294.1255	(M-H)-
18.428	505.2594	251.1072, 343.1382, 459.2162	C31 H38 O6	Amoritin	506.2666	(M-H)-
18.505	193.0868	178.0625, 193.0867	C11 H14 O3	3-Hydroxyphenyl-valeric acid	194.0941	(M-H)-
18.804	415.1765	177.0919, 263.1070, 369.1708	C23 H28 O7	10-Hydroxymelleolide	416.184	(M-H)-
18.916	361.1633	71.0138, 293.1768, 361.1623	C16 H22 N6 O4	Thyrotropin releasing hormone	362.1707	(M-H)-
18.936	293.176	71.014, 236.1059, 293.1762	C17 H26 O4	Myrsinone	294.1833	(M-H)-
19.173	547.2699	59.0141, 147.0448, 263.1077, 395.1645, 455.1865, 547.2698	C33 H40 O7	(+)-Myristinin A	548.2771	(M-H)-
19.384	401.1977	177.0917, 263.1079, 355.1918	C23 H30 O6	Cortisone acetate	402.2049	(M-H)-
19.389	355.192	177.0921, 263.1076, 309.1497	C22 H28 O4	gamma-Crocetin	356.1993	(M-H)-
19.473	194.0824	108.0214, 178.0503	C10 H13 N O3	n-Acetyldopamine	195.0897	(M-H)-
20.29	239.0674	123.0328, 239.0671	C10 H12 N2 O5	Dinoterb	240.0747	(M-H)-

**Table 6 ijms-25-10869-t006:** Analysis of phytochemical constituents of the methanol extract of galangal rhizome (GRhM) performed via LC-QTOF-MS-MS.

RT (min)	*m*/*z*	MS/MS Fragments	Formula	Tentative Identification	Mass	Ion Species
1.279	341.1090	89.0242, 101.024, 341.1086	C12 H22 O11	Sucrose	342.1164	(M-H)-
15.950	311.1140	149.0608, 311.1120	C15 H20 O7	Nivalenol	312.1211	(M-H)-
16.306	341.1240	71.0140, 133.0661, 341.1231	C16 H22 O8	Myzodendrone	342.1317	(M-H)-
17.509	313.1450	112.9855, 175.0399, 251.1082	C19 H22 O4	Gibberellin A120	314.1522	(M-H)-
17.516	359.1500	251.1075, 313.1433	C20 H24 O6	Molephantinin	360.1575	(M-H)-
18.064	279.1030	173.0608, 279.1026	C18 H16 O3	Phenprocoumon	280.1103	(M-H)-
18.173	293.1190	119.0505, 187.0763, 293.1183	C19 H18 O3	(2-Butylbenzofuran-3-yl) (4-hydroxyphenyl)ketone	294.1259	(M-H)-
18.475	355.1550	59.0139, 131.0498, 251.1058, 355.1525	C21 H24 O5	Tephrowatsin C	356.1626	(M-H)-
18.483	401.1610	131.0518, 263.1071, 355.1555	C22 H26 O7	Gmelinol	402.1684	(M-H)-
18.546	327.1600	263.1079, 295.1341	C20 H24 O4	Sagequinone methide A	328.1677	(M-H)-
18.569	373.1660	163.0779, 263.1069, 327.1605	C21 H26 O6	2-(4-Allyl-2,6-dimethoxyphenoxy)-1-(4-hydroxy-3-methoxyphenyl)-1-propanol	374.1733	(M-H)-
18.841	321.1137	173.0596, 279.1028	C20 H18 O4	Neobavaisoflavone	322.1210	(M-H)-
18.913	361.1630	71.0146, 236.1053, 361.1630	C16 H22 N6 O4	Thyrotropin releasing hormone	362.1707	(M-H)-
18.934	293.1762	71.0141, 236.1056, 293.1751	C17 H26 O4	Myrsinone	294.1835	(M-H)-
19.977	265.1480	96.9603, 265.1485	C12 H26 O4 S	Lauryl hydrogen sulfate	266.1557	(M-H)-
20.305	239.0670	151.0757, 207.0409, 239.0666	C10 H12 N2 O5	Dinoterb	240.0747	(M-H)-
20.744	311.1690	119.0503, 183.0123, 311.1682	C20 H24 O3	Triptophenolide	312.1759	(M-H)-
20.917	311.1690	119.0497, 183.0124, 311.1684	C17 H28 O3 S	N-Undecylbenzenesulfonic acid	312.1761	(M-H)-
21.776	293.1800	96.9607, 293.1796	C14 H30 O4 S	Sodium tetradecyl sulfate	294.1871	(M-H)-
21.887	325.1850	79.9579, 183.0126, 325.1828	C18 H30 O3 S	2-Dodecylbenzenesulfonic acid	326.1920	(M-H)-

## Data Availability

Data are contained within the article.

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
