# Peer review of "In Vitro Cytotoxicity and Antimicrobial Activity against Acne-Causing Bacteria and Phytochemical Analysis of Galangal (Alpinia galanga) and Bitter Ginger (Zingiber zerumbet) Extracts"

_ijms, 2024, doi:10.3390/ijms252010869_

Round 1

Reviewer 1 Report

Comments and Suggestions for Authors

The article " In-Vitro Cytotoxicity, Antimicrobial Activity against 2 Acne-Causing Bacteria and Phytochemical Analysis of  Galangal (Alpinia galanga) and Bitter Ginger (Zingiber  zerumbet) Extracts" addresses an interesting topic, specifically the effect of two plants, Galangal and Bitter Ginger, on bacteria responsible for acne. In the context of increasing bacterial resistance to antibiotics, the effect of these plants on certain bacteria is promising, not only in the cosmetic industry but also beyond. The article is well-structured, but for improvement, I recommend the following:

  1. The figures are difficult to follow and need improvement.
  2. The tables are dense and hard to interpret at first glance, requiring adjustments.
  3. There needs to be a stronger emphasis on the specific mechanisms through which the compounds act on bacteria (particularly those responsible for acne).
  4. A more elaborate statistical analysis is needed to support the conclusions.
  5. The conclusions should synthesize the results with implications for practical applications without including figures.
  6. The similarity index is too high to publish in its current format.

Reviewer 2 Report

Comments and Suggestions for Authors

The document is interesting, but has numerous aspects that must be improved:

The title is too generic and should be composed of the main findings of the study;

Please include the concentrations tested in all in vitro tests cited in the abstract;

In the introduction section, the authors could better state the main contributions of this study. Furthermore, revise the last paragraph of the introduction. The first phrase is confused.

A figure of the studied plants would be appropriate.

A positive control groups is missing in the DPPH assay and antimicrobial activity;

The source of plant material must be provided (the market is not trustworthy because it does not provide information regarding the season of the year when it was harvested, storage, or stability…).

The criteria used to select the extracts that were used in the cytotoxicity assay is not clear. Please, state in the text how the selection was performed.

The authors are invited to revise the description of their data. It is imperative that the rationale for selecting the extract for further analysis be clear. The best extract must be identified in each test. The same limitation is observed in the discussion section. A proper discussion linking the findings with the extraction methodology and plant part is necessary.

I recognize that the authors conducted an interesting study, but the findings were not properly explored or explained. In general, the discussion repeats the result section. I also understand that there is a huge amount of data to process and correlate, but the authors should improve the results presentation and discussion.

Considering the application on the skin, other safety assessments should be performed (HET-CAM test).

Please prepare a consistent discussion rationally exploring and correlating the findings instead of repeating the results. Additionally, the authors are invited to include a paragraph of limitations regarding the results and the perspectives.

After reading the manuscript, I found that this study's main contributions and novelties remain unclear. Please state the contributions of this study more clearly.

Comments on the Quality of English Language

There are a few typographical errors in the text.

Reviewer 3 Report

Comments and Suggestions for Authors

Dear Authors,

I write you in regard to your manuscript "In-Vitro Cytotoxicity, Antimicrobial Activity against Acne-Causing Bacteria and Phytochemical Analysis of Galangal (Alpinia galanga) and Bitter Ginger (Zingiber zerumbet) Extracts".

- please, use botanical names in the Material/Methods section.

- items about the characterization of the extracts must be in the sequence of the extract preparation.

- why was Figure 9 after the conclusions?

- it was unclear the DPPH assay value of the control, the vitamin C.

- why was the MIC assay performed? Why not the time-kill test?

- what were the MIC assay controls?

- details of the cytotoxic assay must be provided.

- please, revise the sequence of the items in the manuscript.

Overall, add the results of the controls to compare with the extracts.

Round 2

Reviewer 1 Report

Comments and Suggestions for Authors

The article "In-Vitro Cytotoxicity, Antimicrobial Activity against Acne-Causing Bacteria and Phytochemical Analysis of Galangal (Alpinia galanga) and Bitter Ginger (Zingiber zerumbet) Extracts" has been improved, but it has a similarity index that is too high to be published; this needs to be reduced below 20%. I recommend the authors use a relevant verification program before resubmitting it to avoid complicating the review process.

Reviewer 2 Report

Comments and Suggestions for Authors

I thank the authors for the revision. My concerns were accordingly addressed. 

Reviewer 3 Report

Comments and Suggestions for Authors

Dear Authors,

Thank you for the robust improvements of your manuscript. The term in vitro does not have hyphen.
